# Downregulation of Circulating Hsa-miR-200c-3p Correlates with Dyslipidemia in Patients with Stable Coronary Artery Disease

**DOI:** 10.3390/ijms24021112

**Published:** 2023-01-06

**Authors:** Chiara Vancheri, Elena Morini, Francesca Romana Prandi, Francesco Barillà, Francesco Romeo, Giuseppe Novelli, Francesca Amati

**Affiliations:** 1Unit of Medical Genetics, Department of Biomedicine and Prevention, University of Rome “Tor Vergata”, 00133 Rome, Italy; 2Unit of Cardiology, University Hospital “Tor Vergata”, 00133 Rome, Italy; 3Faculty of Medicine, Unicamillus-Saint Camillus International University of Health and Medical Sciences, 00131 Rome, Italy; 4Neuromed IRCCS Institute, 86077 Pozzilli, Italy; 5Department of Pharmacology, School of Medicine, University of Nevada, Reno, NV 89557, USA

**Keywords:** hsa-miR-200c-3p, genomic biomarker, coronary artery disease, genome-wide methylation, miRNA-sequencing

## Abstract

Coronary heart disease (CHD), one of the leading causes of disability and death worldwide, is a multifactorial disease whose early diagnosis is demanding. Thus, biomarkers predicting the occurrence of this pathology are of great importance from a clinical and therapeutic standpoint. By means of a pilot study on peripheral blood cells (PBMCs) of subjects with no coronary lesions (CTR; n = 2) and patients with stable CAD (CAD; n = 2), we revealed 61 differentially methylated regions (DMRs) (18 promoter regions, 24 genes and 19 CpG islands) and 14.997 differentially methylated single CpG sites (DMCs) in CAD patients. MiRNA-seq results displayed a peculiar miRNAs profile in CAD patients with 18 upregulated and 32 downregulated miRNAs (FC ≥ ±1.5, *p* ≤ 0.05). An integrated analysis of genome-wide DNA methylation and miRNA-seq results indicated a significant downregulation of hsa-miR-200c-3p (FC_CAD_ = −2.97, *p* ≤ 0.05) associated to the hypermethylation of two sites (genomic coordinates: chr12:7073122-7073122 and chr12:7072599-7072599) located intragenic to the miR-200c/141 genomic locus (encoding hsa-miR-200c-3p) (*p*-value = 0.009) in CAD patients. We extended the hsa-miR-200c-3p expression study in a larger cohort (CAD = 72, CTR = 24), confirming its reduced expression level in CAD patients (FC_CAD_ = −2; *p* = 0.02). However, when we analyzed the methylation status of the two CpG sites in the same cohort, we failed to identify significant differences. A ROC curve analysis showed good performance of hsa-miR-200c-3p expression level (AUC = 0.65; *p* = 0.02) in distinguishing CAD from CTR. Moreover, we found a significant positive correlation between hsa-miR-200c-3p expression and creatinine clearance (R^2^ = 0.212, *p* < 0.005, Pearson r = 0.461) in CAD patients. Finally, a phenotypic correlation performed in the CAD group revealed lower hsa-miR-200c-3p expression levels in CAD patients affected by dyslipidemia (+DLP, n = 58) (*p* < 0.01). These results indicate hsa-miR-200c-3p as potential epi-biomarker for the diagnosis and clinical progression of CAD and highlight the importance of deeper studies on the expression of this miRNA to understand its functional role in coronary artery disease development.

## 1. Introduction

Cardiovascular diseases (CVDs) are the leading cause of disease burden and mortality and a major contributor to disability worldwide [1,2].

Coronary artery disease (CAD), the most common form of cardiovascular disease causing more than 3.9 million deaths in Europe [3], is characterized by lipid and inflammatory cell deposits in the inner layer of the coronary arteries. CAD is a multifactorial disease with genetic, epigenetic and acquired risk factors implicated in its etiology.

Dyslipidemia has a key role in the pathogenesis of atherosclerosis, although atherosclerosis does not result simply from a passive accumulation of lipids within the artery wall [4]. Dyslipidemia is associated not only with incorrect plasma lipid concentrations, but also with the presence of dysfunctional lipoproteins. In fact, the evaluation of the lipid profile routinely determined in clinical practice provides only some information about the state of a patient’s lipid metabolism. Lipoproteins can undergo different modifications in vivo that alter their pro- or anti-atherogenic profile (oxidation, glycation, desialylation, carbamylation, nitration and chlorination [5,6]). Further studies based on the molecular mechanisms of lipid modifications, as well as high-quality large-scale human clinical trials are required.

More than 200 autosomal genetic risk loci have been already associated with CAD. A recent genome-wide association study (GWAS) performed on a quarter of a million cases, including cohorts of white, black and Hispanic individuals, identified 95 new loci reaching genome-wide significance [7]. Moreover, the authors found 15 loci of genome-wide significance that robustly overlap with established loci for clinical CAD [7]. These genetic loci are implicated in pathways involved in blood vessel morphogenesis, lipid metabolism, nitric oxide signaling and inflammation [8]. The genetic loci identified so far are extremely important for the implementation of polygenic risk scores (PRS) [9]. In fact, PRS use in clinical practice largely depends on the accuracy of predicting the magnitude of the effect of risk alleles, which varies depending on the genetic background [10].

Most of these CAD genetic risk variations are located in intergenic regions, at or near the promoters, indicating a possible influence on gene expression by epigenetic regulation. Epigenetic mechanisms influence changes in gene expression and function that might be mitotically and/or meiotically heritable and do not entail a change in DNA sequence [11]. They are represented by DNA methylation, histone post-translational modifications and noncoding RNA (ncRNA). In general, the epigenetic modifications are plastic and responsive to external stimuli such as malnutrition, ultraviolet radiation and cigarette smoke [12]. Epigenetic modifications represent a molecular link between environmental factors and regulatory mechanisms of DNA expression.

Epigenetic modifications emerged as new potential processes related to the development of CAD in humans and animal models [13,14] and can also affect CAD risk factors, including atherosclerosis, inflammation, hypertension [15] and diabetes [16].

DNA methylation is one of the earliest epigenetic modifications identified in humans, and it was explored in the context of genomic imprinting [17,18]. The main function of DNA methylation is to modulate gene expression by modifying the accessibility of DNA to the transcriptional machinery [17]. Changes in DNA methylation both in the whole genome and/or in specific a gene site have also been described in CAD, although their specific function is still to be largely investigated [19,20]. Studies reporting a global DNA hypomethylation in advanced atherosclerotic lesions of humans, mice and rabbits indicate also prominent gene clusters linked to hypomethylation in the genomic loci 9p21 and 14q32. However, other studies report a global DNA hypermethylation associated with coronary atherosclerosis (reviewed in [20]). A progressive increase in methylation level was observed in genes involved in macrophage functions and immunity, fundamental aspects of atherosclerosis progression. Genome-wide shotgun bisulfite sequencing revealed marked hypermethylation in the proximity of genes participating in the biology of endothelial and smooth muscle cells, fundamental players in atherogenesis (reviewed in [20]).

MicroRNAs are small noncoding RNAs (on average 18–24 nt in length) extremely conserved from an evolutionary point of view; they regulate gene expression at the post-transcriptional level by binding the 3′-untraslated region (UTR) of specific target mRNA sequences (usually a conserved ~7–8 nucleotide “seed sequence”), thus leading to reduced protein expression by blocking mRNA translation or by promoting mRNA degradation [21,22].

Many studies have shown that microRNAs are involved in the regulation of the development of atherosclerotic plaque and disease progression (reviewed in [20]). A discrete number of miRNAs regulating cholesterol homeostasis (miR-122, miR-223, miR-27b, miR-148a, MiR-33a and miR-33b) have been linked to the promotion of atherosclerosis (reviewed in [20]). These miRNAs are a critical regulator of cholesterol and fatty acid homeostasis, lipoprotein metabolism, negative regulator of LDLR expression or regulator of the HDL/reverse cholesterol transport pathway. Other miRNAs have been identified as relevant players in endothelial dysfunction, vascular smooth cell activation, inflammation and immune response, all key steps of atherosclerosis (reviewed in [20]).

The strong correlations between blood miRNA levels and miRNA expression in the target tissue during disease or injury indicate that miRNAs could be potential non-invasive biomarkers [23,24,25].

The development of new high-throughput methods is now providing researchers with tools to design high-resolution maps of DNA methylation level and miRNA expression in normal tissues versus those in diseases.

By means both genome-wide DNA methylation and miRNAome analysis, we identified a new epigenetic biomarker in coronary artery disease. We profiled miRNA expression and DNA methylation in peripheral blood mononuclear cells (PBMCs) of patients with stable coronary artery disease (CAD group) compared to that of subjects without angiographically significant coronary artery stenosis (controls subjects, CTR group). Our combined analysis revealed that hsa-miR-200c-3p is differentially expressed between CAD patients and control subjects. Our findings make the way to the use of hsa-miR-200c-3p as a potential epi-biomarker for the identification of patients with stable coronary artery disease.

## 2. Results

### 2.1. Clinical Data

The main clinical data of the 96 subjects studied are summarized and reported in Table 1; a more detailed description is present in our previous study [26]. All the enrolled CAD and CTR patients are of white race/ethnicity.

### 2.2. Genome-Wide DNA Methylation Study

A genome-wide DNA methylation study was carried out on two selected patients from each group (Table 2) using the NextSeq500 platform (Illumina; ICM-Plateforme de Génotypage-Séquençage”, Hôpital de la PitiéRNA Salpêtrière, Paris, France).

A hierarchical cluster analysis (HCA), which analyzed differentially methylated regions (DMRs) that comprise promoter regions, genes and CpG islands and differentially single CpG sites (DMCs) in these patients, allowed detecting distinctly different methylation patterns (Figure 1).

CAD patients displayed a higher level of DNA methylation in PBMCs than CTR subjects did (Figure 1).

In order to highlight the DMRs between each group of patients, we analyzed the methylation status of promoters, genes and single sites. In CAD patients, we observed a major level of hypermethylation for promoters, genes, CpG Islands and CpG single sites compared to that in CTR (*p*-value ≤ 0.05 and ΔMeth > 20%). Indeed, CAD patients showed 94% of promoters, 75% of genes, 89% of CpG islands (Figure 2A) and 70% of CpG single sites hypermethylated (Figure 2B).

### 2.3. miRNA-Sequencing Data

miRNA-seq was performed on mRNA extracted from PBMCs of two selected patients of each study group (CTR and CAD group; Table 2) using the NextSeq500 platform (Illumina). By performing an HCA that consistently detected differential expression of several significant miRNAs (*p* < 0.05) among analyzed the patients we distinguished a distinct miRNAs profile in patients featuring the same clinical data. Interestingly, CAD patients showed a miRNAs expression level lower than that of CTR (Figure 3). We detected a common miRNAs signature with which it was worthwhile to distinguish the group of patients from the group of controls (Figure 3).

Considering a *p*-value ≤ 0.05 and a fold-change (FC) ≥ ±1.5, we identified in the CAD group 50 differential regulated miRNAs (18 upregulated and 32 downregulated; Appendix A).

### 2.4. Integrated Analysis of miRNA-Sequencing and Genome-Wide Methylation Results

Then, we performed an integrated analysis between miRNA-seq and genome-wide methylation data. In fact, we aimed to identify further biomarkers potentially useful to discriminate CAD patients from individuals with healthy coronary arteries (CTR).

Three interactions between miRNAs and DMCs were statistically significant (*p* ≤ 0.05) in CAD patients compared with those in CTR (Table 3). Table 3 reports the differentially expressed miRNAs with the change/regulation in terms of methylation of the corresponding miRNA gene.

If we examine each miRNA’s expression level and the methylation status of the single CpG site located in the corresponding miRNA gene, the expression of hsa-miR-200c-3p and hsa-miR-655-3p could be correlated with the methylation status of the single CpG site inside the corresponding miRNA’s genes (Table 3, in italic). In fact, these miRNAs were downregulated, while the CpG single sites, mapping in the miRNA precursor regions, were hypermethylated.

We selected hsa-miR-200c-3p, since this miRNA showed the higher statistically significant (*p* = 0.009) downregulation in CAD patients compared to that in control subjects (FC = −2.97). Moreover, analyzing the methylation status, it was the only miRNA that showed two differential methylated sites (genomic coordinates: chr12:7073122-7073122 and chr12:7072599-7072599) inside its genomic locus, miR-200c/141. Finally, it is known for its involvement in cardiovascular diseases.

### 2.5. Validation of the Integrated miRNA-Sequencing and Genome-Wide Methylation Results on Four Selected Patients

In order to confirm the integrated miRNA-sequencing and genome-wide methylation results, we performed an hsa-miR-200c-3p expression analysis by qRT-PCR on the four patients (Table 2) selected for these studies, and a methylation assay by Pyrosequencing of the two DMS (genomic coordinates: chr12:7073122-7073122 and chr12:7072599-7072599) that are intragenic in the miR-200c/141 genomic locus. Both analyses confirmed the results obtained by the miRNA-seq and genome-wide methylation studies. Indeed, we observed lower expression of hsa-miR-200c-3p in CAD patients compared to that in the controls (*p* < 0.05) (Figure 4A) and hypermethylation status of chr12:7073122-7073122 (Figure 4B, *p* < 0.005) and chr12:7072599-7072599 (Figure 4C, *p* < 0.005) sites in the same CAD patients compared to that in the same CTR subjects.

### 2.6. Validation of Integrated miRNA-Sequencing and Genome-Wide Methylation Results in All-Case Study

After the first validation step with the four subjects analyzed for miRNA-seq and DNA methylation, we repeated the same analyses on all recruited CAD patients and CTR subjects.

Interestingly, the validation of the expression level of this miRNA in the whole recruited sample confirmed the reduced expression in CAD patients compared to that in the controls (FC = −2; *p* < 0.05) (Figure 5).

Moreover, to investigate the potential use of hsa-miR-200c-3p as a diagnostic and prognostic biomarker for CAD patients, receiver operator characteristic (ROC) analyses were performed. An ROC curve of hsa-miR-200c-3p reflects a statistically significant separation between patients. Indeed, hsa-miR-200c-3p has high power for the discrimination of CAD patients from CTR subjects (AUC 0.65, 95% confidence interval 0.53–0.77, *p* < 0.05) (Figure 6).

On the contrary, the analysis of the methylation level of the two different sites (genomic coordinates: chr12:7073122-7073122 and chr12:7072599-7072599) located intragenic to the miR-200c/141 genomic locus (encoding hsa-miR-200c-3p) on all recruited CAD and CTR patients did not show a statistically significant difference (Figure 7).

### 2.7. Correlation Analysis

We analyzed by a Pearson correlation test the relationship between the expression level of hsa-miR-200c-3p and the available clinical data of our case study. Regression analyses showed a significant positive correlation between hsa-miR-200c-3p and creatinine clearance levels in the CAD group (Figure 8).

In addition, a phenotypic correlation performed in the CAD group showed that CAD patients affected by dyslipidemia (+DLP, n = 58) displayed lower hsa-miR-200c-3p expression levels when compared to those of CAD patients not affected by dyslipidemia (-DLP, n = 7) (*p* < 0.01) (Figure 9).

## 3. Discussion

CAD, the most important and widespread cardiovascular disease, is a multifactorial disease to whose development contribute genetic, epigenetic and environmental risk factors [1]. Over the years, genetic studies identified hundreds of genetic variants responsible for the increased risk of developing coronary artery disease and myocardial infarction. However, these loci explain about 15% of CAD hereditability, are carried mainly on European populations and are autosomal. Recently, a GWAS conducted on a quarter of a million cases, including cohorts of white, black and Hispanic individuals, identified 95 novel genetic susceptibility loci associated with CAD. Moreover, these authors demonstrated that the two common haplotypes at the 9p21 locus, which are associated with a ~30% increased risk of CAD per copy of the risk allele, are responsible for risk stratification in all populations except those of African origin, in which these haplotypes are virtually absent [7].

In this context, the epigenetic mechanisms are acquiring an increasingly important role to identify possible biomarkers for the risk stratification in cardiovascular diseases. Indeed, recently epigenetic alterations in DNA methylation, histone modifications, miRNAs and long noncoding RNAs (lncRNAs) expression have been related to the basis of the complex pathogenesis of CAD [20,27]. In fact, during the last years, increasing evidence showed that epigenetic modifications affect the onset and the risk to develop CAD [16,19,20].

This pilot study is aimed to identify potential epigenetic biomarkers for risk stratification of CAD patients through a miRNA-sequencing and genome-wide methylation study.

The miRNA-sequencing study, performed on four selected subjects (two CTR and two CAD) revealed several differentially expressed microRNAs (Figure 3). In addition, the genome-wide DNA methylation analysis, performed on the same selected patients, determined a qualitative profile of methylation characteristics of CAD patients and CTR subjects (Figure 1) and revealed a high number of hypermethylated DMRs and DMCs in CAD patients compared to those in the controls. This global DNA hypermethylation result is in agreement with previously reported data in the literature [28,29].

After an integration analysis of miRNA-seq and genome-wide methylation results, we identified hsa-miR200c-3p as interesting for our analysis in an all-case study (n = 96), as it was downregulated in CAD patients (n = 72; *p* < 0.05) compared to control subjects (n = 24) (Figure 5). In addition, we observed a positive association among the expression level of hsa-miR-200c-3p and the methylation status of single CpG sites located inside the corresponding miRNA gene (Table 3, in italic). Noteworthy, the significant downregulation of this miRNA may be correlated with the hypermethylation of the CpG single sites mapping in the miRNA’s precursor regions. Interestingly, in the miRNA’s precursor region of hsa-miR-200c-3p two differentially methylated single CpG sites are mapped (chr12:7073122 ∆Meth% = 31%, *p* ≤ 0.05 and chr12:7072599 ∆Meth% = 24%, *p* ≤ 0.05) (Table 3). Considering the known involvement of hsa-miR200c-3p in the mechanisms underlying coronary artery disease [30] and given its downregulation in PBMCs of CAD patients compared to that in controls (Figure 5), we analyzed the methylation status of these two sites on all recruited subjects, but we did not observe a statistically significant difference (Figure 7). This result may indicate that the expression level of hsa-miR-200c-3p in PBMCs of CAD patients does not depend exclusively on the methylation levels of these two putative regulatory sites. In addition, the individual variability in our case study could play an important role in the miRNA’s expression study.

A ROC curve analysis showed that this microRNA has high discriminatory power and thus is an excellent candidate as a new epigenetic biomarker for risk stratification of CAD patients (Figure 6). hsa-miR-200c-3p is one of the five members of the hsa-miR-200 family (hsa-miR-200c and hsa-miR-141 clustered on chromosome 12, and hsa-miR-200a, hsa-miR-200b and hsa-miR-429 clustered on chromosome 1). This miRNA family displays characteristic features in vascular cell response to oxidative stress [31]. Moreover, hsa-miR-200c-3p is involved in different processes related to the onset of cardiovascular risk factors (diabetes mellitus and obesity) and to the development of atherosclerosis [31]. In addition, hsa-miR-200c-3p is one of the most abundant miRNAs present in cardiomyocyte-derived extracellular vesicles that mediate functional cross-talk with endothelial cells [32]. Finally, as reported by Chen et al., hsa-miR-200c-3p has been implicated in epithelial-to-mesenchymal transition (EMT) [33], thus suggesting a probable involvement of this miRNA in the formation of cardiomyocytes, cardiac fibroblasts and cardiac fibers undergoing the EMT process.

Interestingly, regression analyses indicate a significant positive correlation among hsa-miR-200c-3p and creatinine clearance values in CAD group (Figure 8). Creatinine clearance represents an approximate measure of glomerular filtration rate (GFR) that describes the flow rate of fluid filtered through the kidneys. Creatinine clearance is a rapid and cost-effective method for the measurement of renal function since the glomerulus freely filters creatinine [34], although this method slightly overestimates GFR by 10–20% because creatinine is moderately secreted by renal tubules [35]. Creatinine is a waste product generated by creatine phosphate (or phosphocreatine) breakdown in skeletal muscle and it plays a key role in cellular energy metabolism. Indeed, phosphocreatine acts as an energy reserve to synthesize ATP rapidly, with no need for oxygen. This reaction plays a crucial role in heart contraction [36,37]. It is estimated that >98% of creatinine comes from the muscle, where it is produced and secreted into the serum at a continuous rate [37,38]. Once in the serum, creatinine is almost exclusively excreted in the urine. Indeed, elevations in serum creatinine typically reflect decrements in glomerular filtration rate (GFR) [38,39]. However, the relationship between serum creatinine and GFR is nonlinear, as serum creatinine typically begins to rise significantly as GFR declines to <60 mL/min. Therefore, when GFR is in the normal range, relatively large decreases in GFR result in only small increases in serum creatinine; conversely, when the GFR is significantly decreased, small decrements in GFR produce relatively large changes in serum creatinine [40]. Noteworthy, patients with low muscle mass and low meat intake have lower GFR for any given creatinine level. Decreased GFR is a key component of the development of accelerated atherosclerosis, perhaps mediated by impaired clearance of pro-atherogenic cytokines and hormones by the kidney [41].

In CAD patients, we observed downregulation of hsa-miR-200c-3p compared to that in controls, and a positive correlation among hsa-miR-200c-3p levels and creatinine clearance values, suggesting that CAD patients exhibited reduced creatinine clearance values compared to those in healthy subjects. This result could be related to the concomitant presence of chronic kidney disease (CKD). The prevalence of traditional cardiovascular risk factors such as diabetes, hypertension and hyperlipidemia are very high in patients with CKD. In addition, several uremia-related risk factors such as inflammation, oxidative stress, endothelial dysfunction, coronary artery calcification and hyperhomocysteinemia have been associated with accelerated atherosclerosis in these patients [42]. The presence of CKD is an independent risk factor for poor outcomes, including myocardial infarction and cardiovascular death, in patients with CAD. Joachim et al. showed that a lower creatinine excretion rate (CER) is strongly associated with mortality in patients with CAD, independent of conventional measures of body composition and traditional CAD risk factors [43,44].

These findings suggest that hsa-miR-200c-3p expression levels may be related to CAD disease progression and may also have a prognostic role in CAD patients.

Finally, we performed a phenotypic correlation in the CAD group and observed that patients with dyslipidemia (+DLP) showed lower hsa-miR-200c-3p levels (hsa-miR-200c-3p mean level CAD +DLP = 0.03) when compared to those of CAD patients without DLP (hsa-miR-200c-3p mean level CAD − DLP = 0.25) (Figure 9). Dyslipidemia, including elevated blood levels of low-density lipoprotein cholesterol (LDL-C), total cholesterol (TC), tryglicerides or reduced levels of high-density lipoproteins (HDL), represents a prominent risk factor for CVD and CAD [45]. Several epidemiological and Mendelian randomization studies together with randomized controlled trials demonstrate a log-linear relationship between the absolute changes in plasma LDL-C and the risk of atherosclerotic cardiovascular disease (ASCVD) [46]. Oxidized LDL retained in the arterial wall represents the initial “injury” that, through interaction with the endothelial lectin-type ox-LDL receptor 1 (LOX-1), activates the inflammatory “response to injury”, with involvement of innate and adaptive immunity, that is crucial in the atherosclerotic plaque formation process [47,48]. Several studies provide evidence that individual circulating miRNAs are associated with dyslipidemia and CAD [49]. In a recent study, Li et al. showed that hsa-miR-200c-3p is involved in pyroptosis (a highly inflammatory programmed cell death characterized by cell rupture, nuclear formation, and high concentrations of nuclear serous fluid) by targeting SLC30A7 in patients affected by diabetic retinopathy (DR) [50]. SLC30A7 is involved in the regulation of insulin secretion and in the regulation of diabetic cardiomyopathy progression by influencing muco-mitochondrial coupling in hyperglycemic cardiomyocytes [51,52].

## 4. Materials and Methods

### 4.1. Patient Recruitment and Sample Collection

We studied a total of 96 patients, already enrolled for another study, including 72 patients with coronary artery disease (CAD group) and 24 control subjects with free coronary arteries as the control group (CTR group). The CTR group included subjects without angiographically significant coronary artery stenosis. The principal clinical characteristics of these patients are summarized and reported in Table 1. A more detailed description of clinical data of these patients is available in Vancheri et al. [26].

From each patient, one blood sample (9 mL) was collected within 24 h from the coronary angiography (PROTOCOL DOI: dx.doi.org/10.17504/protocols.io.zpvf5n6, accessed on 31 December 2022). Biochemical analyses including a complete blood count, basic metabolic profile (glucose, HbA1c, renal function tests, liver function tests) and lipid profile (total cholesterol, LDL, HDL, triglycerides) were performed. Creatinine serum (mg/dL) and creatinine clearance (mL/min) calculated with the Cockcroft–Gault formula were also detected.

This project was approved by the Ethical Committee of Policlinico Tor Vergata-Fondazione PTV (n. 30/15). All the principles outlined in the Helsinki Declaration of 1975, as revised in 2013, were followed in all the assays involving human subjects during the current study.

### 4.2. DNA Extraction from Whole Blood Samples

Genomic DNA was manually extracted from whole blood samples (300 µL) using a FlexiGene DNA Kit (QIAGEN, https://www.qiagen.com/us, accessed on 31 December 2022) according to the manufacturer’s instructions. The concentration of the isolated DNA was evaluated by using a NanoDrop ND-1000 Spectrophotometer (Euro-Clone, 20016 Pero, Italy).

### 4.3. PBMCs Isolation

PBMCs were isolated by whole blood (8 mL) using Ficoll^®^ Paque Plus (GE Healthcare, https://www.gehealthcare.com/, accessed on 31 December 2022) according to manufacturer instructions. Isolated PBMCs were suspended in 1 mL of Trizol^®^ (Ambion, Waltham, MA, USA) and stored at −80 °C until further analysis.

### 4.4. Total RNA Extraction from PBMCs and Reverse Transcription of miRNAs

Total RNA extraction from PBMCs was performed using Trizol^®^ (Ambion, Waltham, MA, USA) according to manufacturer instructions. RNA concentration was evaluated by using a NanoDrop ND-1000 Spectrophotometer (Euro-Clone, 20016 Pero, Italy), whereas RNA quality was checked on 1% agarose gel. To isolate miRNAs fraction, 50 ng of RNA was reverse-transcribed into cDNA using a miScript II RT kit (QIAGEN, https://www.qiagen.com/us, accessed on 31 December 2022) following the manufacturer’s instruction.

### 4.5. Genome-Wide Methylation Study and miRNA Sequencing

Genome-wide methylation and miRNA sequencing (miRNA-seq) studies were performed in collaboration with “ICM-Plateforme de Génotypage-Séquençage” (Hôpital de la Pitié Salpêtrière, Paris, France) and GenoSplice (www.genosplice.com, accessed on 31 December 2022) on two selected patients from CTR and CAD groups, matched for age and clinical features accordingly with their medical condition, already analyzed in another study (Table 2) [26].

A genome-wide methylation study was performed on genomic bisulfite-converted DNA. For sequencing, performed using a NextSeq 500 ILLUMINA platform, 500 ng of fresh bisulfite-treated DNA input was used, while for library preparation, Kapa ROCHE together with CpGiant ROCHE Nimblegen was used. GenoSplice (www.genosplice.com, accessed on 31 December 2022) performed the quality control and further analyses of BS-Seq data. All samples passed quality control analysis and after quality trimming of paired end reads with Trimgalore v0.4.1, the reads were mapped against hg19 reference sequence using Bismark v0.16.1 with Bowtie2. Bismark_deduplicate (Bismark v0.16.1) was used in order to remove PCR duplication artifacts. Then, the methylation status was extracted with Bismark_methylation_extractor (Bismark v0.16.1). Finally, the differentially methylated analysis was performed using the rn-beads R packages for gene, promoter, CpG island and site levels. Clustering analysis and heat maps were performed using “dist” and “hclust” functions in R, Euclidean distance and the Ward agglomeration method.

For miRNA sequencing, total RNA (500 ng) was used as the input material. Sequencing libraries were generated using SMARTer small RNA CLONTECH for Illumina following the manufacturer’s recommendations (New England BioLabs, Ipswich, MA, USA). Index codes were added to attribute sequences to each sample. All samples passed the quality control analysis.

MiRNAs data analysis and sequencing data quality check was performed using FastQC. Sequences were trimmed by cutadapt (v0.9.5) before analysis. MiRNAs proportion in the sequences was evaluated using Bowtie (v0.12.7) for mapping sequences on the hg19 Human genome assembly and using HTSeq to count reads per gene on GeneCode v19 annotations. MicroRNAs expression was obtained by mapping and quantifying sequences with mirDeep2 on mirbase v21. Based on these read counts, normalization and differential miRNA expression were performed using edgeR on R (v.3.2.5) on samples paired by patient. Only miRNAs with at least 5 mapped reads were considered as expressed. Results were considered statistically significant for *p*-values ≤ 0.05 and fold-changes ≥±1.5. Whole-genome DNA methylation and RNA-seq raw data were submitted to GEO with the record GSE209779.

### 4.6. Integrated Analysis of miRNA-Sequencing and Genome-Wide Methylation Results

MiRNA data integration analysis was conducted by using miRTarBase v6.0. This new database describes miRNA–target interactions for different species, based on interactions annotated as “Functional MTI (Moving Target Indication)”. The interaction reported in miRTarBase was based on experimentally validated MTIs from the literature [53].

### 4.7. Methylation Analysis of Selected CpG Sites Differentially Methylated (DMCs) by Pyrosequencing

The methylation study on two selected CpG sites differentially methylated (DMCs), both intergenic to the miR-200c/141 locus on chromosome 12 encoding miR-200c was conducted on bisulphite-treated DNA before Pyrosequencing analysis.

The two sites were: 1. chr12:7072599-7072599 (GRCh37-hg19), a CpG island with 5 CpG sites; 2. chr12:7073122-7073122 (GRCh37-hg19) which is a single CpG site. To convert all unmethylated cytosines of genomic DNA to uracil, a bisuphite treatment was performed on 2 µg of genomic DNA using an EZ DNA Methylation-Gold Kit (Zymo Research, Irvine, CA, USA) according to the manufacturer’s instructions. The genomic treated DNA was quantified by a Qubit^®^2.0 Fluorometer (ThermoFisher Scientific, https://www.thermofisher.com, accessed on 31 December 2022).

For site 1, a 130 bp fragment was amplified by PCR using the following primers: Chr12:707255-F (5′-GGGATGAGGGTGGGTAAAT-3′) and Chr12:7072599-R (5′-biotinylated-ACCCAAATTACAATCCAAACAAA-3′). For site 2, a 104 bp fragment was amplified using the following primers: Chr12:7073122-F (5′-GCCACTAAGGGACACAATGGG-3′) and Chr12:7073122-R (5′-biotinylated-TTTGAGTTTGGGGTTGGTTC-3′). PCR conditions were 94 °C for 10 min, followed by 50 cycles of 94 °C for 40 s, 58 °C for 40 s and 68 °C for 1 min with a final extension of 10 min at 72 °C. Biotinylated PCR products were then processed using a PyroMark Q24 Vacuum Workstation (QIAGEN), and subsequent pyrosequencing was performed on a PyroMark Q24 pyrosequencer using PyroMark Gold Q24 Reagents (QIAGEN, https://www.qiagen.com/us, accessed on 31 December 2022) [54] with the following sequencing primers: Chr12:7072599-F for site 1 and Chr12:7073122-F for site 2. The methylation percentage at each CpG region was quantified using PyroMark Q24 software, version 2.0.7 (QIAGEN, https://www.qiagen.com/us, accessed on 31 December 2022).

### 4.8. MiRNA-Specific Expression by Quantitative Real-Time PCR

For the hsa-miR-200c-3p expression study, we applied previously described experimental protocols [23]. The expression level of hsa-miR-200c-3p was analyzed on all recruited patients (24 CTR subjects and 72 CAD patients) by qRT-PCR using a miScript SYBR Green PCR kit (QIAGEN, https://www.qiagen.com/us, accessed on 31 December 2022) and a miRNA-specific miScript primer assay (QIAGEN). RNA U6 was used for data normalization and analysis of the results since it showed stable expression among our samples. Data analysis was performed using the comparative threshold cycle (Ct) method quantification (2-ΔCT method) (PROTOCOL DOI: dx.doi.org/10.17504/protocols.io.zp7f5rn; accessed on 31 December 2022).

### 4.9. Statistical Analysis

Statistical analysis was performed using GraphPad Prism 6.0 (GraphPad Software, San Diego, CA, USA) and SPSS, version 19 (IBM Corp, Amonk, NY, USA).

The Kolmogorov–Smirnov test was used to analyze the distribution of experimental data (i.e., miRNA expression data) for each analyzed group (CAD and CTRL). The Mann–Whitney test is a nonparametric statistical test used to compare two samples or groups. Student’s t-test was used to determine significant difference between two groups when assuming that these two populations were normally distributed. For parametric and nonparametric distribution, expression data are represented as the mean and standard deviation (SD).

The receiver operating characteristic (ROC) curve was used to determine the specificity of the candidate miRNA biomarkers to discriminate between the two groups of patients. For all analysis, significance was set at *p* ≤ 0.05.

## 5. Conclusions

We are aware that even if our patient and control cohorts were carefully selected, the heterogeneous composition of different blood cells within whole blood samples can influence the global DNA methylation level [55,56,57,58]. In addition, the possibility that DNA sequence variations (i.e., SNPs) can interfere with DNA methylation profiling must be evaluated [59]. Indeed, the presence of SNPs in CG nucleotides may alter the DNA methylation results. Thus, further studies with uniformity in material and measurement techniques are needed to make statements that are more conclusive. Moreover, as it appears from recent genetic studies [7], the evaluation of diverse populations might contribute to better characterization of the epigenetic architecture of CAD.

Taken together, our results show that PBMCs of CAD patients present distinct and specific DNA methylation and miRNAs profiles, i.e., a higher level of DNA methylation and lower miRNA expression. These profiles might be considered characteristic of their medical conditions and provide novel insights in the pathogenesis of this complex disease.

Our whole-genome methylation and miRNA-sequencing study has led to the identification of a new potential epigenetic biomarker, hsa-miR-200c-3p, for CAD patients. We showed that miR-200c-3p expression level demonstrated good discriminatory ability for risk stratification in stable CAD patients when compared to control subjects. Based on these data and also on published data describing miR-200c as a promising target to manipulate epicardial cell fate and, potentially, cardiac repair and regeneration [60], future studies on the expression levels of this miR in CAD patients after a myocardial event (MI) will be desirable.

Finally, we noticed differential expression of hsa-miR-200c-3p in dyslipidemic and non-dyslipidemic patients that could suggest its potential role in altering the amount of circulating lipids that have a key role in the pathogenesis of atherosclerosis and a causal effect on the risk of atherosclerotic cardiovascular diseases. The differential expression of hsa-miR-200c-3p in dyslipidemic and non-dyslipidemic patients could suggest the potential role of this miRNA in altering the amount of circulating lipids, particularly LDL-C, which was shown to have a key role in the pathogenesis of atherosclerosis and a causal effect on the risk of atherosclerotic cardiovascular diseases. Therefore, hsa-miR-200c-3p might have a role in the development of accelerated atherosclerosis and CAD disease progression.

This pilot study confirms the need of deeper investigations on miR-200c-3p as a new epigenetic biomarker with diagnostic and, hopefully in the future, prognostic value.

## Figures and Tables

**Figure 1 ijms-24-01112-f001:**
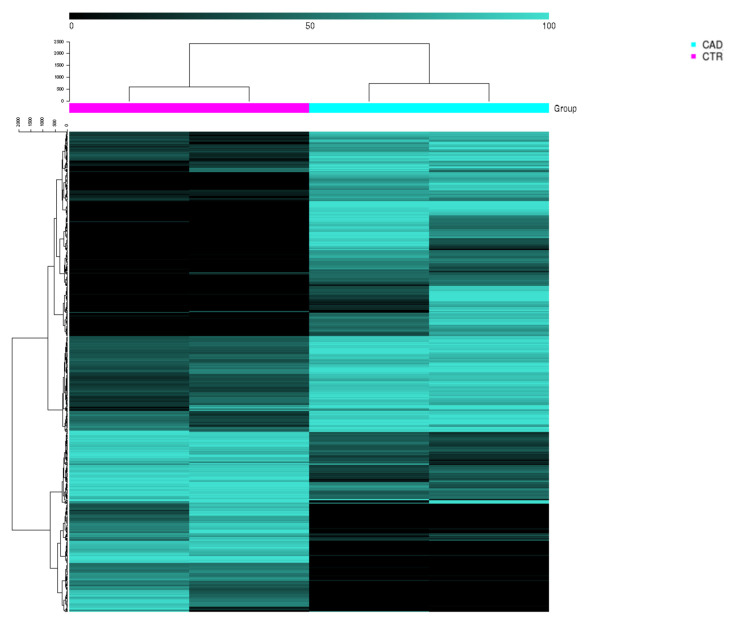
HCA analysis and heat maps of genome-wide methylation study. Graphical representation of the single sites differentially methylated (DMCs). Black represents hypomethylated genomic regions (0% of methylation), while light green—the hypermethylated ones (100% of methylation). Only the DMCs with ΔMeth > 50% have been depicted.

**Figure 2 ijms-24-01112-f002:**
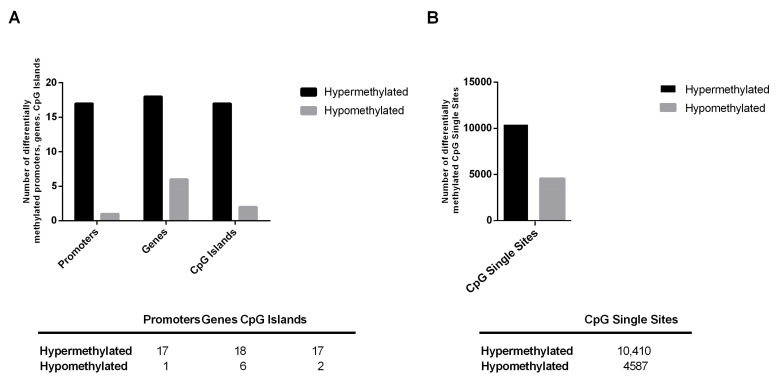
Methylation status in the comparison of CAD vs CTR. (**A**) Number of differentially methylated regions (DMRs) as promoters, genes and CpG Islands. (**B**) Number of differentially methylated CpG single sites (DMCs). *p* ≤ 0.05, ΔMeth > 20%.

**Figure 3 ijms-24-01112-f003:**
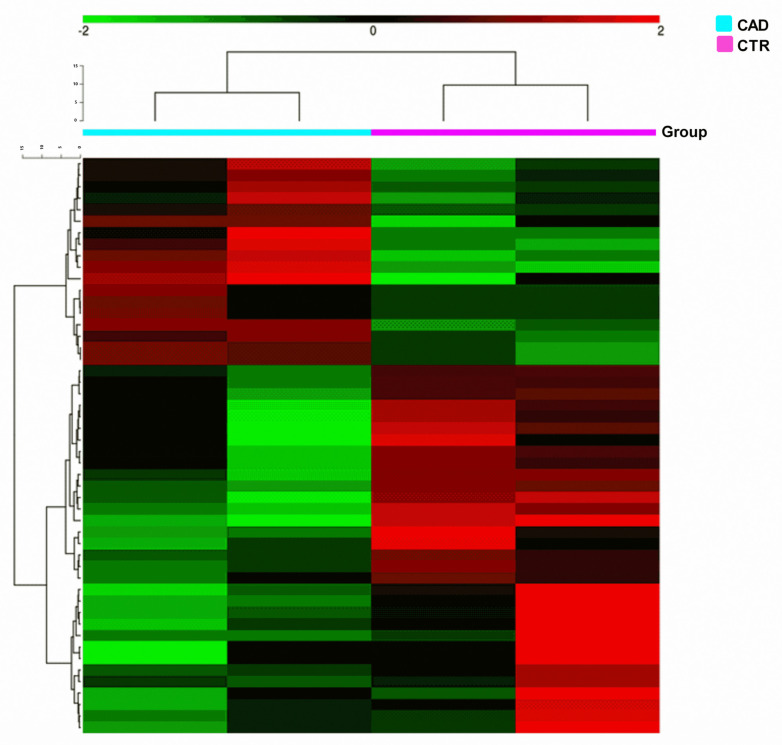
Heat maps of miRNA sequencing study. Green represents miRNAs with an expression level under the mean, while red shows miRNAs with an expression level above the mean.

**Figure 4 ijms-24-01112-f004:**
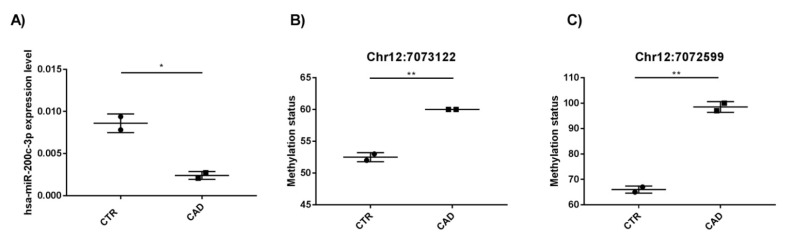
Integrated miRNA-sequencing and genome-wide methylation results on four selected subjects (2 CTR and 2 CAD). (**A**) Expression level of hsa-miR-200c-3p; (**B**) methylation status of site chr12:7073122-7073122; (**C**) methylation status of site chr12:7072599-7072599. * *p*-value < 0.05; ** *p*-value < 0.005.

**Figure 5 ijms-24-01112-f005:**
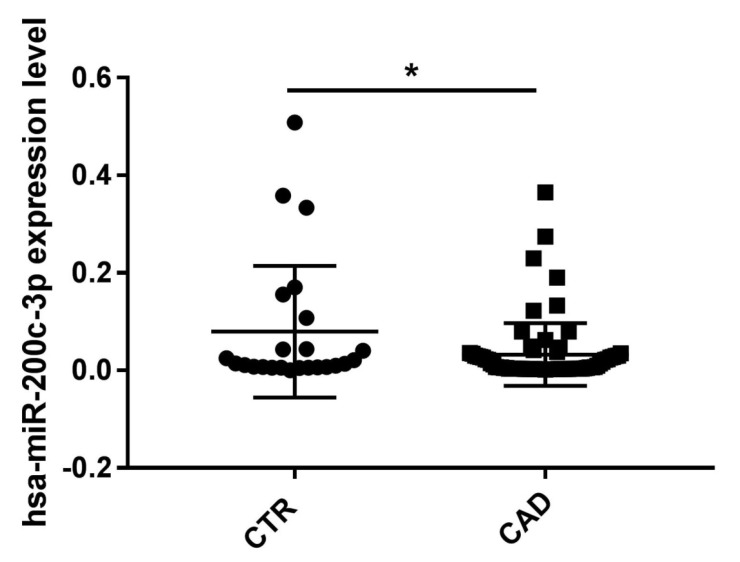
Expression level of hsa-miR-200c-3p in the comparison CAD vs CTR patients. Mann–Whitney test. * *p*-value < 0.05. Expression data are represented as the mean and SD.

**Figure 6 ijms-24-01112-f006:**
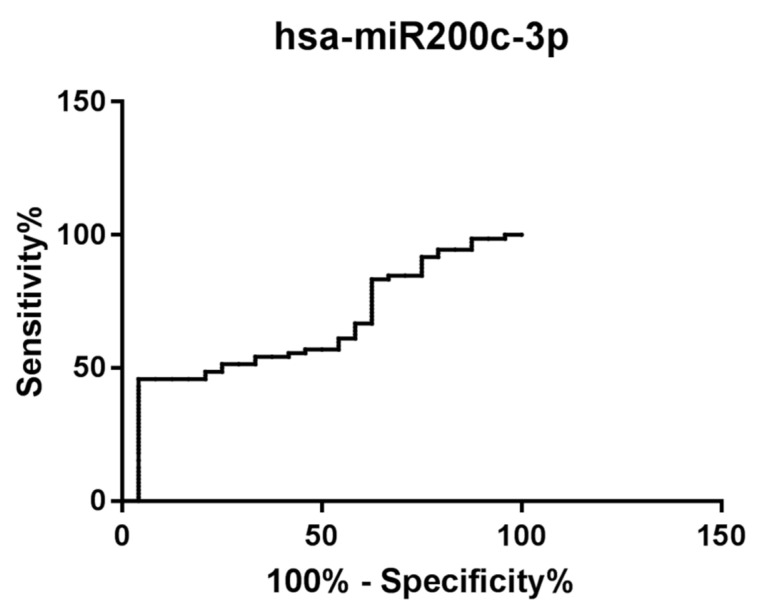
Receiver operator characteristic (ROC) analysis of hsa-miR-200c-3p expression levels. Area under the ROC curve (AUC) = 0.65, 95% confidence interval: 0.53 to 0.77, *p*-value < 0.05.

**Figure 7 ijms-24-01112-f007:**
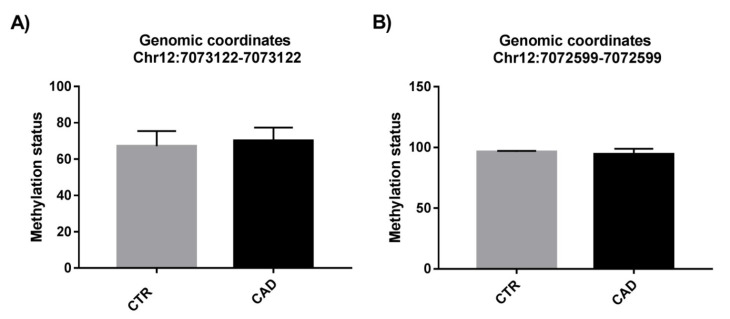
Methylation status of the two sites on all recruited CAD and CTR patients. (**A**) Methylation level for site chr12:7073122-7073122; (**B**) methylation level for site chr12:7072599-7072599.

**Figure 8 ijms-24-01112-f008:**
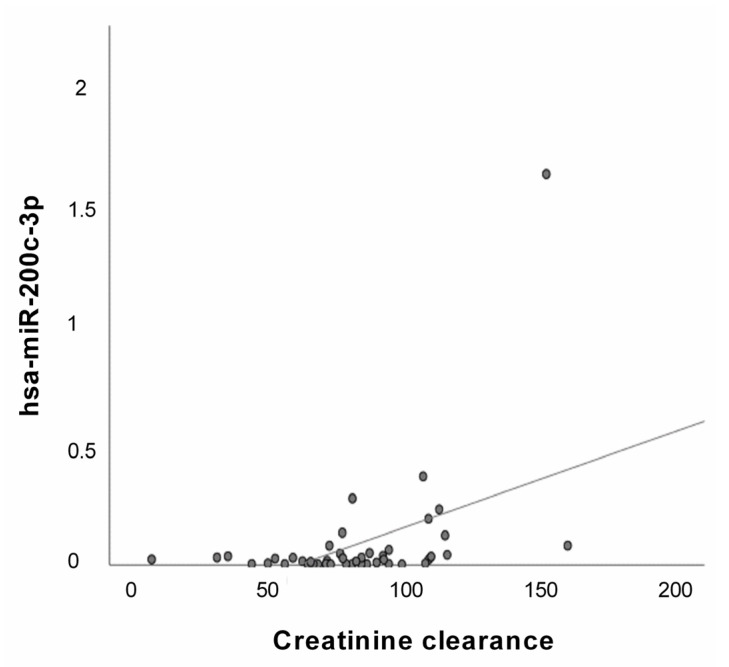
Correlation analysis of has-miR-200c-3p in the CAD group. Scatter plots depict the relationship among hsa-miR-200c-3p and creatinine clearance values in the CAD group (R^2^ = 0.212, *p* < 0.005, Pearson r = 0.461).

**Figure 9 ijms-24-01112-f009:**
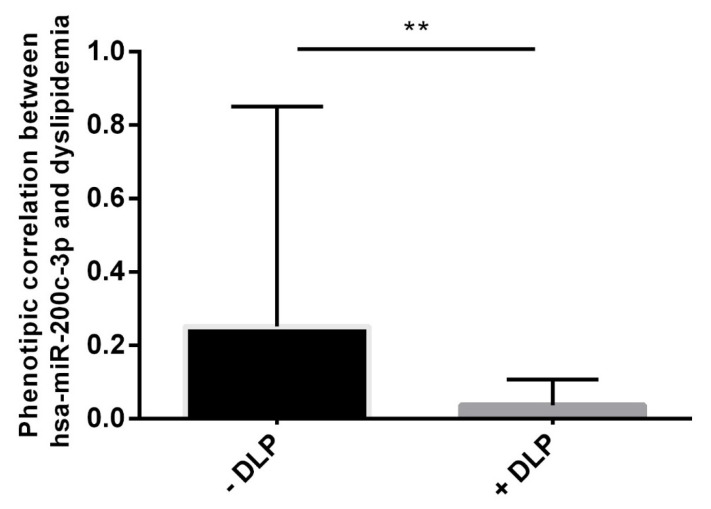
Phenotypic correlation of has-miR-200c-3p and dyslipidemia in the CAD group. -DLP: CAD not affected by dyslipidemia. +DLP: CAD affected by dyslipidemia. *t*-test. ** *p*-value < 0.01. Expression data are represented as the mean and SD.

**Table 1 ijms-24-01112-t001:** Clinical features of CTR subjects and CAD patients studied.

	CTR Subjects	CAD Patients	*p*-Value
Age (years)	67.5 ± 9.3	66.6 ± 9.8	n.s.
Gender			
Male (%)	66.6	84.7	*p*-value < 0.05
Hypertension (%)	69.6	77.5	n.s.
Diabetes (%)	21.7	39.4	n.s.
Dyslipidemia (%)	47.8	88.7	*p*-value < 0.0005
Smoking history			
Present (%)	17.4	23.9	n.s.
Past (%)	43.5	39.4	n.s.
Number of affected vessels			
1 vessel disease (%)	0	45.1	
2 vessel disease (%)	0	29.6	
3 vessel disease (%)	0	25.4	
Type of affected vessel			
LAD (%)	0	54.9	
CFX (%)	0	31	
RCA (%)	0	38	

Continuous data are expressed as the mean ± SD; categorical data are expressed as a percentage. LAD, left descending artery; CFX, circumflex coronary artery; RCA, right coronary artery; 0 = absence. Student’s *t*-test was used to assess significance. n.s. = not significant.

**Table 2 ijms-24-01112-t002:** Clinical features of CTR subjects and CAD patients (n = 2) selected for miRNA-sequencing and genome-wide methylation studies.

	CTR Subjects (n = 2)	CAD Patients (n = 2)	*p*-Value
Age (years)	70.5 ± 11.1	71.5 ± 4.9	n.s.
Gender			
Male (%)	100	100	n.s.
Hypertension (%)	100	100	n.s.
Diabetes (%)	0	50	n.s.
Dyslipidemia (%)	100	100	n.s.
Smoking history			
Present (%)	0	0	n.s.
Past (%)	0	100	*p*-value < 0.05
Number of affected vessels			
1 vessel disease (%)	0	50	n.s
2 vessel disease (%)	0	0	n.s.
3 vessel disease (%)	0	50	n.s.
Type of affected vessel			
LAD (%)	0	50	n.s.
CFX (%)	0	100	*p*-value < 0.05
RCA (%)	0	50	n.s.

Continuous data are expressed as the mean ± SD; categorical data are expressed as a percentage. LAD, left descending artery; CFX, circumflex coronary artery; RCA, right coronary artery; Student’s *t*-test was used to assess significance. 100 = both patients manifested the clinical sign. 50 = only one patient showed the clinical sign. 0 = neither patient had the clinical sign. n.s. = not significant.

**Table 3 ijms-24-01112-t003:** Integrated analysis between miRNAs and methylation data in the comparison of CAD vs CTR patients.

miRNAs					Methylation			
**miR Coordinates**	**hsa-miR ID**	**Regulation**	**FC**	* **p** * **-Value**	**DMSs**	**Genomic Coordinates (GRCh38)**	**Status**	* **p** * **-Value**
*chr12:7072362-7073429*	*miR-200c-3p*	*down*	*−2.97*	*0.009*	*2*	*chr12:7073122-7073122* *chr12:7072599-7072599*	*Hypermethylated*	*0.024* *0.046*
*chr14:101515387-* *101516483*	*miR-655-3p*	*down*	*−3.6*	*0.016*	*1*	*chr14:101515591-101515591*	*Hypermethylated*	*0.004*
chr7:1062069-1063162	miR-339-5p	up	2.15	0.041	1	chr7:1063097-1063097	Hypermethylated	0.017

In italic, the miRNAs and their corresponding single differentially methylated single CpG sites (DMCs) were characterized by an inverse trend. DMSs: differential methylated sites. FC: fold change.

## Data Availability

Whole-genome DNA methylation and RNA-seq raw data have been submitted to GEO with the record GSE209779.

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
