# Peer review of "Downregulation of Circulating Hsa-miR-200c-3p Correlates with Dyslipidemia in Patients with Stable Coronary Artery Disease"

_ijms, 2023, doi:10.3390/ijms24021112_

Round 1
Reviewer 1 Report
This study is a pilot one on two patients with coronary artery disease and two controls in order to find a biomarker for predicting the occurrence of the disease. To do so, methylation status of 61 regions in promoters, genes, and CpG islands as well as single CpG sites were detected. Also, some differentially expressed miRNAs were identified between patients and control groups. The authors reported that down regulation of miR-200c-3p was associated to the hypermethylation of two sites in its promoter regions. The study in interesting and merits consideration. I invite the authors to respond to the following criticisms:
- Introduction
Although harboring useful information, the introduction section seems long as it contains certain facts about the prevalence of CAD and its importance, risk factors, genetic susceptibility, and extra explanations about epigenetic regulation. Instead, I believe that the authors replace these with some justifications regarding the relevance of epigenetic changes and miRNA expression, which is one of the major aspect of the study.
- Methods
I didn't understand why you report 96 subjects while 90% of your findings were on two CAD and two CTR groups.
- Results
Table 1: Describing p value in some variables like severity of the disease (1-vessel, 2-vessel, …) that we are aware about the status of the control group (being negative) is useless.
Table 2: Number of patients in each group is not clear. It is better to use yes/no instead of percent to describe their characteristics.
Fig 2: How did you define the status of the control group in the charts? Do you assume zero for the control group?
Line 156: What does it mean by a peculiar profile in those with same clinical data? Does it mean that miRNA profile is different among CAD patients (or control group)? Make it clear.
Lines 162: 18 upregulated miRNAs are in the control group?
Line 164: In CAD group or CTR group?
Table 3: Please determine that the p value is the resultant of which comparisons. Table should be completed by extra explanations on the parameters that are mentioned in the table.
Line 187: What is the relevance of having two differentially methylated sites for choosing the miRNA of interest?
- Discussion
Line 273: It is better to include the race/ethnicity of your participants as it seems important to your discussion where this subject was mentioned in the first paragraph of this section.
Lines 285-6: in only two CAD subjects? If it is right, include it.
Reviewer 2 Report
The authors found that hsa-miR-200c-3p may serve as a potential epi-biomarker for the diagnosis and clinical progression of CAD.
The study is well designed, and the results are well presented and discussed.
Major comment: Could the authors comment or have data concerning the role of has-miR-200c-3p on the progress of disease?
Reviewer 3 Report
We suggest that the "material and method" section to be placed before the "result" session and to add a paragraph of conclusions
Reviewer 4 Report
I have received to review the research article entitled “Downregulation of circulating hsa-miR-200c-3p correlates with dyslipidemia in patients with stable coronary artery disease”, prepared by Chiara Vancheri et al. submitted to the International Journal of Molecular Sciences (IF=6.208). Cardiovascular diseases (CVDs) in the course of atherosclerosis are the leading cause of morbidity and mortality worldwide. Dyslipidemia is an important risk factor for CVDs development. Research related to increasing the knowledge about pathogenesis of CVDs are therefore of crucial importance.
In my opinion, the presented paper is generally well prepared and it represents high scientific level. This paper should be considered for publication, but some revisions are necessary and I listed it below.
1) Although the introduction is informative and correctly written, it should be shortened and only the most important information on genetics and epigenetics should be elucidated. Basic information on dyslipidemia and its role in the pathogenesis of atherosclerosis should be highlighted. It should be noted that dyslipidemia is associated not only with incorrect plasma lipid concentrations, but also with the presence of dysfunctional lipoproteins. (10.3390/antiox11050856)
2) In the table 2. the absence should be marked also as “0” (similarly to the table 1).
3) I believe that the description of the statistical analysis is laconic and should be supplemented. It should be precisely described for what purpose each test was used. The t-test should not be used to test for "clinical differences" because one should not speak of clinical significance in the context of statistical analysis, but only of statistical significance.
4) What biochemical tests were performed on the study participants, what were their results and what methodology was used? From the text of the paper it can be concluded that creatinine clearance was studied, but it is not directly described in the methodology.
5) At the end of the work, it is worth posting a short summary of the results obtained.
Round 2
Reviewer 4 Report
The paper has been improved significantly. I have no further comments.